# Rapid and Quantitative Determination of Sildenafil in Cocktail Based on Surface Enhanced Raman Spectroscopy

**DOI:** 10.3390/molecules24091790

**Published:** 2019-05-09

**Authors:** Lei Lin, Fangfang Qu, Pengcheng Nie, Hui Zhang, Bingquan Chu, Yong He

**Affiliations:** 1College of Biosystems Engineering and Food Science, Zhejiang University, Hangzhou 310058, China; linlei2016@zju.edu.cn (L.L.); ffqu@zju.edu.cn (F.Q.); npc2012@zju.edu.cn (P.N.); 13644410041@163.com (H.Z.); bqchu@zust.edu.cn (B.C.); 2Key Laboratory of Spectroscopy Sensing, Ministry of Agriculture, Zhejiang University, Hangzhou 310058, China; 3State Key Laboratory of Modern Optical Instrumentation, Zhejiang University, Hangzhou 310058, China; 4School of Biological and Chemical Engineering, Zhejiang University of Science and Technology, Hangzhou 310023, China

**Keywords:** sildenafil, cocktail, surface enhanced Raman spectroscopy, quantitative determination, partial least squares

## Abstract

Sildenafil (SD) and its related compounds are the most common adulterants found in herbal preparations used as sexual enhancer or man’s virility products. However, the abuse of SD threatens human health such as through headache, back pain, rhinitis, etc. Therefore, it is important to accurately detect the presence of SD in alcoholic beverages. In this study, the Opto Trace Raman 202 (OTR 202) was used as a surface-enhanced Raman spectroscopy (SERS) active colloids to detect SD. The results demonstrated that the limit of detection (LOD) of SD was found to be as low as 0.1 mg/L. Moreover, 1235, 1401, 1530, and 1584 cm^−1^ could be qualitatively determined as SD characteristic peaks. In a practical application, SD in cocktail could be easily detected using SERS based on OTR 202. Also, there was a good linear correlation between the intensity of Raman peaks at 1235, 1401, 1530, and 1584 cm^−1^ and the logarithm of SD concentration in cocktail was in the range of 0.1–10 mg/L (0.9822 < R^2^ < 0.9860). The relative standard deviation (RSD) was less than 12.7% and the recovery ranged from 93.0%–105.8%. Moreover, the original 500–1700 cm^−1^ SERS spectra were pretreated and the partial least squares (PLS) was applied to establish the prediction model between SERS spectra and SD content in cocktail and the highest determination coefficient (Rp^2^) reached 0.9856. In summary, the SD in cocktail could be rapidly and quantitatively determined by SERS, which was beneficial to provide a rapid and accurate scheme for the detection of SD in alcoholic beverages.

## 1. Introduction

Sildenafil (SD) and its related compounds are the most common adulterants found in herbal preparation, which can be used as sexual enhancer or man’s virility products [1]. Its pharmacological effect is to inhibit the metabolism of the second messenger cyclic guanosinc monophosphate (cGMP), promote the relaxation of cavernous artery smooth muscle, and then improve the symptoms of erectile dysfunction (ED) [2,3]. However, the usage of SD is controlled through medical supervision due to their harmful side-effects such as headache, dyspepsia, back pain, rhinitis, flu syndrome, etc. [4]. In recent years, SD, through illegal business, has illegally added to Chinese patent medicines and alcoholic drinks in pursuit of high profits. Generally, traditional methods for determining SD in alcoholic beverages include ultraviolet spectrophotometry (UV) [5,6], high-performance liquid chromatography (HPLC) [7], gas chromatography-mass spectrometry (GC–MS) [8,9,10], thin-layer chromatography (TLC) [11,12], and near Infrared Spectrometry (NIR) [13,14]. Xin et al. [15] applied HPLC method to detect seven kidney-tonifying and yang-strengthening illegal additives. It was shown that the limits of detection (LODs) were in the range of 8.2–33.4 ng. Kee et al. [16] achieved the classification of two sets of isomers (sildenafil analogues and mercaptosidinafil analogues) by Orbitrap mass spectrometry (MS) method, and the database of these compounds was established using quality processing software. Tagami et al. [17] established a method for the isolation and identification of five sildenafil analogues using liquid chromatography-mass spectrometry (LC-MS). Huang et al. [18] achieved the qualitative and quantitative detection of 17 kidney-tonifying and yang-strengthening chemicals that were illegally added in Chinese patent medicine by LC-MS method. The results suggested that the LODs were in the range of 0.05–1.25 mg/kg. Although these detection methods achieved high sensibility, the developments of these methods were limited by the cumbersome pre-test, time-consuming detection, inconvenient instrument, expensive reagents, and other shortcomings [19].

Compared with the methods mentioned above, surface-enhanced Raman spectroscopy (SERS), which can provide ultrasensitive and unmarked chemical analysis, has attracted focus and attention in past decades [20,21]. Besides, SERS is suitable for rapid screening of molecule substances’ absorbed molecules because of its advantages of simple pretreatment, convenient equipment, and fast detection speed [22]. In the field of SD analysis and detection using SERS technique, scholars have carried out relevant researches. Liu et al. [12] applied the thin-layer chromatographic surface- enhanced Raman spectroscopy (TLC-SERS) to detect the SD in proprietary medicines and health products. It was found that 1563, 1530, 1405, 1240, 1272, and 1582 cm^−1^ could be qualitatively determined as SD characteristic peaks, while the quantitative detection of SD was not involved. Zhen et al. [23] combined SERS with gold and silver nanoparticles to analyze SD in health wine. It was indicated that the LOD of SD reached 0.5 mg/kg. However, the correlation coefficient (R^2^ = 0.9472) was not high in the range of 0.1 to 10 mg/L. Wang et al. [24] found that 11 kinds of SD drugs could be classified into five groups according to their structures. The results demonstrated that the LOD of the SD was found to be as low as 0.05 mg/kg using SERS technology. Although the linear correlation coefficient (R^2^ = 0.97) was relatively high, there were only five samples in the linear regression equation. Yu et al. [25] achieved the rapid detection of SD citrate in health wine using disperse magnetic solid phase microextraction and surface enhanced Raman scattering (Dis-MSPME-SERS). The results suggested that the minimum detectable concentration was 1.0 × 10^−8^ M. But the Dis-MSPME-SERS pretreatment method was complex and time-consuming.

Above the analysis, it is important to accurately detect SD in health wine using SERS. In this paper, the SD in cocktail was selected as the research object. The main objective of this paper is to rapidly and quantitatively determine SD in cocktail combined with chemometric methods and improve the accuracy of SD detection, which is of great significance for rapid and accurate detection of SD in cocktail.

## 2. Results and Discussion

### 2.1. Opto Trace Raman 202 and its Spectral Analysis

To investigate the enhancement effects of Opto Trace Raman 202 (OTR 202), the structure, UV spectrum, and Raman spectroscopy (RS) of OTR 202 were analyzed. Figure 1a is the transmission electron microscopy (TEM) image of OTR 202, Figure 1b displays the UV/Visible spectra of OTR 202, and Figure 1c shows the Raman spectrum of OTR 202.

It can be clearly seen that the average diameter of OTR 202 was about 30 nm. As shown in Figure 1b, the UV/Visible characteristic absorption peaks of OTR 202 was at 533 nm. Beside this, the Raman spectrum of OTR 202 only had a faint signal at 1630 cm^−1^ (Figure 1c), suggesting that OTR 202 themselves had no strong Raman characteristic peaks and did not have an interferential effect on experimental results. Therefore, the OTR 202 was suitable as SERS substrate to detect SD in this paper.

### 2.2. The SD Molecule and its Assignment of Raman Peaks

The molecular structure of SD (molecular formula: C_22_H_30_N_6_O_4_S) is shown in Figure 2a. Density functional theory (DFT), as a common method for molecular geometry optimization and frequency vibration calculation, can describe the ground state physical properties of atoms and molecules [26]. In this paper, DFT was applied to calculate SD molecule and optimize its structure in Gaussian.v09 software. In this software, the vibrational form of relevant chemical bonds calculated by Hartree-Fock wave function can be obtained [27]. Figure 2b is the RS of SD simulated by DFT, Figure 2c is the RS of solid SD. Besides, in order to verify the necessity of using OTR 202 reinforcement, the SERS of SD was analyzed. Figure 2d shows the SERS spectra of SD with OTR 202.

As seen in Figure 2, the positions of SD spectral bands and its intensities were basically consistent with the experiment-detected Raman spectra of SD (Raman shifts < 5 cm^−1^), which indicated that the position of Raman peaks detected by SRES spectra based on OTR 202 were feasible and reliable. Except for the differences between experiment-detected Raman spectra of SD and the DFT-calculated Raman spectra of SD at 812 and 1487 cm^−1^, the DFT-calculated Raman spectra of SD were basically similar to the experiment-detected Raman spectra of SD whose Raman shifts (less than 10 cm^−1^) were within a reasonable range. Combined with related literature [23], the assignments of Raman peaks of SD are listed in Table 1. For the SERS of SD, 552 cm^−1^ was the carbonyl stretching and phenetole breathing deformable vibration; 647 cm^−1^ was the carbonyl stretching, phenetole deformable vibration, and C-S stretching in sulfamide, and 831 cm^−1^ belonged to the pyrazole pyridine stretching; 922 cm^−1^ was assigned to the C–C deformable vibration and C–H stretching in pyrazole pyridine group, 1235 cm^−1^ was the C–H stretching vibration in carbonyl, 1305 cm^−1^ was C–H stretching vibration in ethyl. Further, 1401 cm^−1^ was the C–H deformable vibration in methyl piperazine.;1488, 1530, and 1584 cm^−1^ were the C–H deformable vibration in pyrazole pyridine.

### 2.3. Limit of Detection of SD

To investigate the sensitivity and stability of the OTR 202 substrates for the detection of SD in cocktail, six different SD concentrations (0, 0.1, 0.5, 2, 5, and 10 mg/L) were collected and the corresponding SERS are shown in Figure 3A.

According to Figure 3A,B, the SERS intensity decreased gradually with the decrease of SD concentration from 10 mg/L to 0 mg/L. It was found that the characteristic peaks at 1235, 1401, 1530, and 1584 cm^−1^ of SD in cocktail were still identified even when the SD solution concentration was as low as 0.1 mg/L. It can be seen that there was faint SERS signal when the SD was 0 mg/L. This might belong to the SERS signal of some substances in cocktails. Generally, the LOD of SD in cocktail reached 0.1 mg/L and 1235, 1401, 1530, and 1584 cm^−1^ could be qualitatively determined as SD characteristic peaks.

### 2.4. Detection of SD in Cocktail

In this study, to investigate the accuracy and stability of the OTR 202 substrate for the detection of SD in cocktail, SD in cocktail with the concentrations ranging from 10 mg/L to 0.1 mg/L were detected using SERS. The SERS spectra of SD in cocktail were obtained, and the representative SERS spectra are given in Figure 4.

As shown in Figure 4, with the decrease of SD in cocktail from 10 mg/L to 2 mg/L, the Raman peaks at 1235, 1401, 1530, and 1584 cm^−1^ decreased. Therefore, 1235, 1401, 1530, and 1584 cm^−1^ could be quantitatively determined as SD characteristic peaks in cocktail. With a portable Raman system, quantitative SERS determination of SD concentration (0.1, 0.2, 0.4, 0.6, 0.8, 1, 2, 4, 6, 8, and 10 mg/L) in cocktail has been conducted and the linear regression equations between Raman peak intensity at 1235, 1401, 1530, and 1584 cm^−1^ and logarithm of SD concentration in cocktail were established, respectively (Figure 5).

According to Figure 5, SERS spectra of SD in cocktail mixed with the OTR 202 are concentration dependent. The peaks at 1235, 1401, 1530, and 1584 cm^−1^ could be regarded as a marker band for SD in cocktail determination owing to its drastic intensity change with varying SD concentration. There was a good linear correlation between Raman peak intensity and logarithm of SD concentration in cocktail in each linear regression equation ranged from 0.1 mg/L to 10 mg/L (0.9822 < R^2^ < 0.9860), which demonstrated that the SERS can accurately and quantitatively analyze SD in cocktail. To verify the accuracy of this method, first, eight different SD concentration in cocktail (0.3, 0.5, 0.7, 0.9, 3, 5, 7, and 9 mg/L) were prepared and each concentration contained three samples. Second, all the samples were detected by SERS based on OTR 202 of three consecutive days. Third, the linear regression equations at 1235, 1401, 1530, and 1584 cm^−1^ were used to predict the SD concentration in cocktail. Table 2 presents the precision and accuracy of method for the determination of sildenafil in cocktail.

According to Table 2, the intra and inter day relative standard deviation (RSD) were less than 12.7% and 11.8%, respectively. The intra and inter day accuracy% ranged between 93.0%–105.6% and 93.6%–105.8%, respectively. From the results, one can infer that precision and accuracy were within the acceptable limits.

### 2.5. Determination of SD in Cocktail with PLS

Considering that the Raman peaks of SD in cocktail were mainly distributed in the range of 500–1700 cm^−1^, the partial least squares (PLS) prediction model was established based on 500–1700 cm^−1^ SERS spectra. The SERS spectra of SD in cocktail of 155 samples were obtained and then pretreated with Savitzky–Golay (S-G), detrend (DT), standard normal variation (SNV), and 1st-derivative (1st-Der), respectively, and then modeled by PLS. The sample set portioning based on the joint x-y distance (SPXY) [28] method was used to separate the soil samples into calibration set and validation set at the ratio of 2:1. The PLS modeling results based on the 500–1700 cm^−1^ spectra under different pretreatments are presented in Table 3. The scatter plot between the predicted values and the measured values of the correction set and the prediction set sample are shown in Figure 6.

It can be seen that the predictive effect of SD in cocktail was great (0.9896 < Rc^2^ < 0.9948, 0.216 < RMSEC < 0.310; 0.9760 < Rp^2^ < 0.9856, 0.354 < RMSEP < 0.434). Moreover, the modeling effect was similar after using different pretreatment methods. Among them, the SERS original spectra performed a slightly better modeling effect compared with SNV and 1st-Der. On the one hand, it was shown that the background noise had little effect on the original spectrum and the PLS model with good effect could be established through the original spectrum. On the other hand, although the SNV and 1st-Der eliminated the influence of SRES background noise to some extent, it might weaken the spectral resolution and made it difficult for quantitative analysis.

## 3. Materials and Methods

### 3.1. Experimental Instruments and Reagents

In this study, the experimental instruments included: (1) RmTracer-200-HS portable Raman spectrometer combined with a 785 nm excitation wavelength diode-stabilized stimulator (Opto Trace Technologies, Inc., Mountain View, CA, USA); (2) The FEI Tecnai G2 F20 S-TWIN transmission electron microscope (FEI Company, Hillsboro, OR, USA); (3) Vortex-Genie 2/2T vortex mixer (Shanghai Ling early Environmental Protection Instrument Co., Ltd., Shanghai, China). Moreover, the experimental reagents included: (1) Sildenafil (99.8% purity, Sigma-Aldrich, Beijing, China); (2) methanol (chromatographically pure, Amethyst Chemicals, Beijing, China); (3) cocktail (Shanghai Baxter Liquor Co., Ltd., Shanghai, China). In addition, the OTR 202 nanomaterials and OTR 103 produced by Opto Trace Technologies, Inc. (SuZhou, China) were also used in this study.

### 3.2. Sample Preparation

The process of specific sample preparation was as follows. First, the standard of SD was diluted to 1000 mg/L with methanol. Second, the standard solution of 1000 mg/L was diluted to 0 to 1 mg/L (0.1 mg/L per gradient, 11 concentrations) and 1.2 to 10 mg/L (0.2 mg/L per gradient, 44 concentrations) with cocktail. There were 3 samples for each concentration. A total of 155 samples were prepared.

### 3.3. SERS Measurement

Before Raman spectra acquisition, the instrument should be calibrated using a 785 nm excitation wavelength. The parameters were set as follows: A power of 200 mw, a scanning range of 200 to 3300 cm^−1^, an optical resolution of 2 cm^−1^, an integration time of 10 s, and an average spectral value of 3 times. The solid SD RS collection was that SD powder was in quartz plate with glass slides flattened and the spectra were acquired with matching microscope platform. When collecting the SERS of samples, 500 μL OTR 202, 100 μL test solution, and 100 μL OTR 103 were added in turn into a 2 mL quartz bottle, then it was placed at a liquid sample pool.

### 3.4. Density Functional Theory (DFT)

Density functional theory (DFT), as a tool for calculating molecular energy and analyzing properties, has been widely used in the field of physics and chemistry. It provides the computational strategies for obtaining information about the energetics, structure, and properties of atoms and molecules [29]. The performance of Becke three-parameter Lee–Yang–Parr (B3LYP) functional in combination with various basis sets has been extensively tested for molecular geometries, vibrational frequencies, ionization energies and electron affinities, dipole and quadrupole moments, atomic charges, infrared intensities, and magnetic properties [27]. Among the various functions and basis sets in DFT, the hybrid functional B3LYP with the 6-31G (d,p) basis set has been commonly used in the Raman spectroscopic calculation of biological molecules [30]. In this paper, B3LYP/6-31G (d,p) was used for the theoretical simulation and calculation of SD molecules.

### 3.5. Spectral Preprocessing Methods

The SERS spectra could be affected by instrument resolution, laser energy, instrument parameters, environmental factors, scattering light on quartz bottle surface, and optical path change. Thus, removing background fluorescence from Raman signals is essential for analyzing Raman signals accurately. Here in the PLS model, each original SERS spectrum was processed by the S-G [31] 5 points smoothing filter, Detrend, SNV, and 1st-Der, respectively. S-G smoothing can reduce the noise introduced by samples, instrument state, surrounding environment, and human operation. The principle of SNV [32] algorithm is that the absorbance values of each wavelength point satisfies a certain distribution in each spectra, and the spectral correction was performed according to this assumption, which can eliminate the influence of scattering light and path change on SERS spectrum of quartz bottle surface in liquid detection. The idea of detrend (DT) [33] algorithm is that the spectral absorbance and wavelength are first fitted into a trend line *d* according to the polynomial, and then the trend line *d* is subtracted from the original spectra *x* to achieve the effect of the trend. 1st-derivation (1st-Der) can distinguish overlapping peaks and eliminate interference from other backgrounds, which improves spectral resolution, sensitivity, and the signal-to-noise ratio of the spectra [34].

### 3.6. Partial Least Squares Model

PLS is a commonly-used calibration model for spectral data analysis, which reflects the relationship between spectra and attribution information due to its flexibility and reliability [35,36]. When PLS is applied to dealing with spectral data, the spectral matrix is decomposed first and the main principal component variables are obtained, then the contribution of each principal component is calculated. The flexibility of PLS makes it able to interpret the dependent and independent variables well by establishing regression models. In this study, the PLS model was established with the SERS spectral data as *X* and the content of SD as *Y*, whose best principal factor was determined by the root mean square error of cross validation (RMSECV). In addition, all above-mentioned data analysis in this study were performed on OMNIC v8.2 (Thermo Nicolet Corp., Madison, WI, USA), MATLAB R2014a (The MathWorks, Inc., Natick, MA, USA), and Gaussian.v09 (Gaussian, Inc., Wallingford, CT, USA).

### 3.7. Model Evaluation Index

In this experiment, the modeling effect was evaluated by the coefficient of determination (R^2^), the root mean square error (RMSE), relative standard deviation (RSD), and recovery rate. The coefficient of determination R^2^ reflects the level of intimacy between variables and the RMSE reflects the model accuracy. The lower the RMSE, and the closer the R^2^ is to 1, the better the performance of the prediction model. In this study, Rc^2^ and Rp^2^ represent the coefficient of the determination of the calibration set and the prediction set respectively, while RMSEC and RMSEP represent the root mean square error of the calibration set and the prediction set, respectively [37,38]. In addition, relative standard deviation (RSD) reflects the degree of discretization between individuals in the reflection group and recovery rate reflects the degree of coincidence between the results of the reaction and the true value. The recovery rate ranges closer to 100%, and the lower the RSD, the better the reliability of the model [39].

## 4. Conclusions

In this paper, we reported the rapid and quantitative determination of SD in cocktail based on SERS with OTR 202. According to the results, we found that there was a good linear correlation between Raman peak intensity at 1235, 1401, 1530, and 1584 cm^−1^ and logarithm of SD concentration in cocktail with R^2^ from 0.9822 to 0.9860 in the range of 0.1–10 mg/L and the LOD could reach 0.1 mg/L. Also, the determination coefficient (Rp^2^) for SD in cocktail in PLS model was great (0.9760 < Rp^2^ < 0.9856). It was indicated that the rapid detection of SD by SERS was feasible and reliable. Overall, the SERS method with OTR 202 enhancement developed through this study provide a novel, rapid, and accurate approach to quantitatively determine SD in cocktail, which could meet the requirements of analysis and detection of SD in other alcoholic beverages.

## Figures and Tables

**Figure 1 molecules-24-01790-f001:**
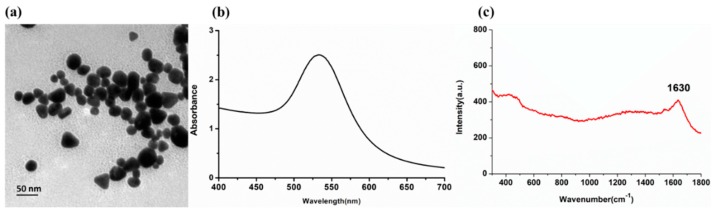
(**a**) Transmission electron microscopy (TEM) images of Opto Trace Raman 202 (OTR 202); (**b**) the UV/Visible spectra of OTRT 202; (**c**) the Raman spectrum (RS) of OTR 202.

**Figure 2 molecules-24-01790-f002:**
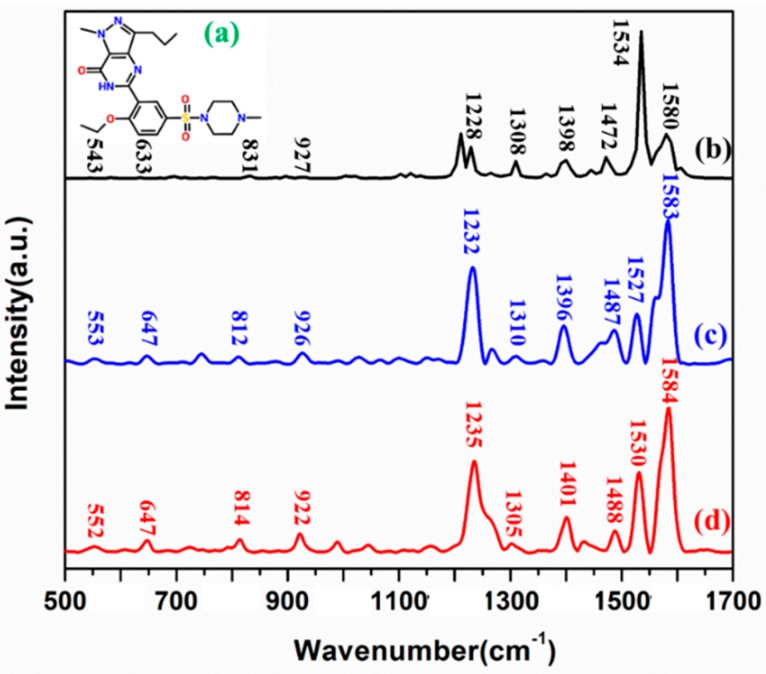
(**a**) Molecular structure of sildenafil (SD); (**b**) the theory calculation by density functional theory (DFT); (**c**) RS of solid SD. (**d**) Surface-enhanced Raman spectroscopy (SERS) spectra of SD.

**Figure 3 molecules-24-01790-f003:**
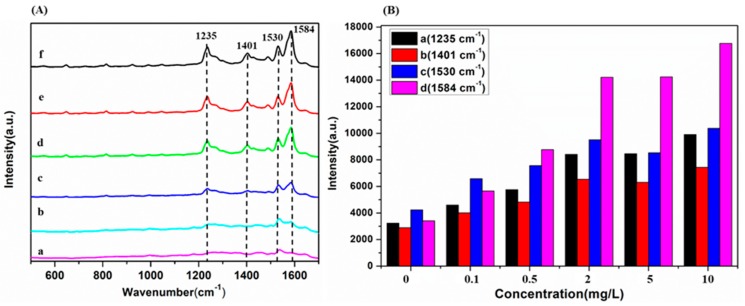
(**A**) The SERS of sildenafil (SD) in cocktail. (**a**) 0 mg/L; (**b**) 0.1 mg/L; (**c**) 0.5 mg/L; (**d**) 2 mg/L; (**e**) 5 mg/L; (**f**) 10 mg/L. (**B**) The SERS intensity of SD concentration from 10 mg/L to 0 mg/L at 1235 cm^−1^, (b) 1401 cm^−1^, (c)1530 cm^−1^ and (d) 1584 cm^−1^.

**Figure 4 molecules-24-01790-f004:**
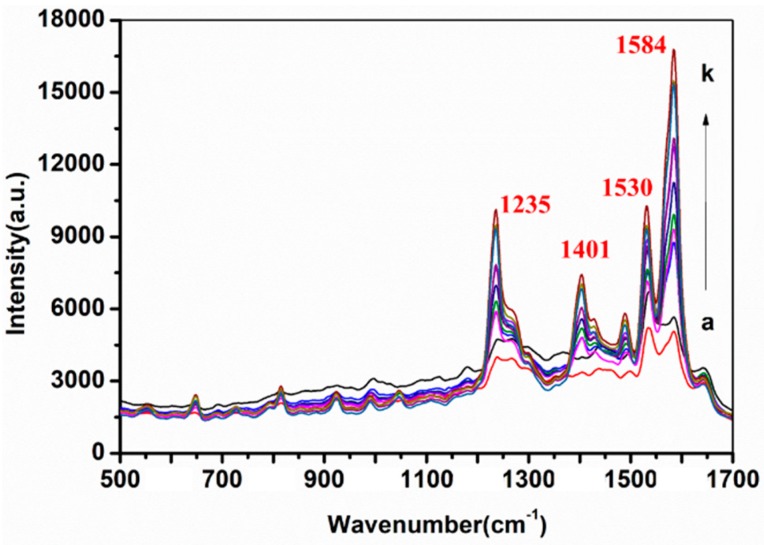
SERS spectra of sildenafil (SD) in cocktail with different concentrations from a to k: 0.1, 0.2, 0.4, 0.6, 0.8, 1, 2, 4, 6, 8, and 10 mg/L, respectively.

**Figure 5 molecules-24-01790-f005:**
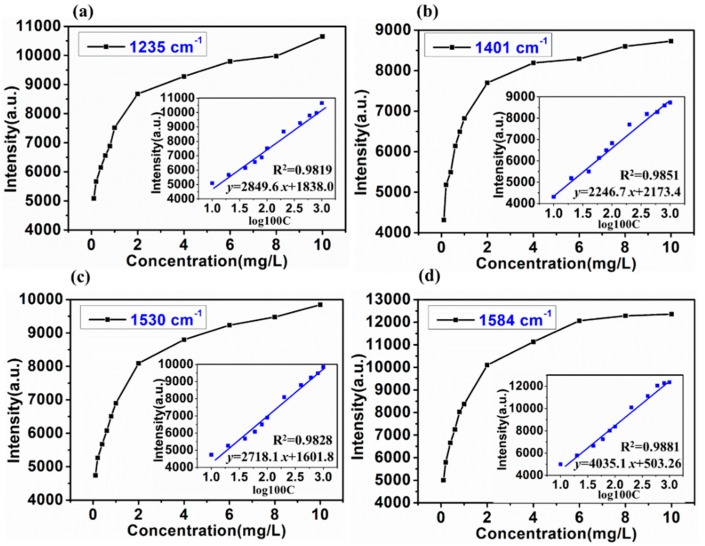
The plot of intensities of SERS peak at 1235 cm^−1^ (**a**), 1401 cm^−1^ (**b**), 1530 cm^−1^ (**c**), and 1584 cm^−1^ (**d**) versus SD concentration in cocktail. Inset: The linear calibration plotted in the logarithm concentration range from 0.1 mg/L to 10 mg/L.

**Figure 6 molecules-24-01790-f006:**
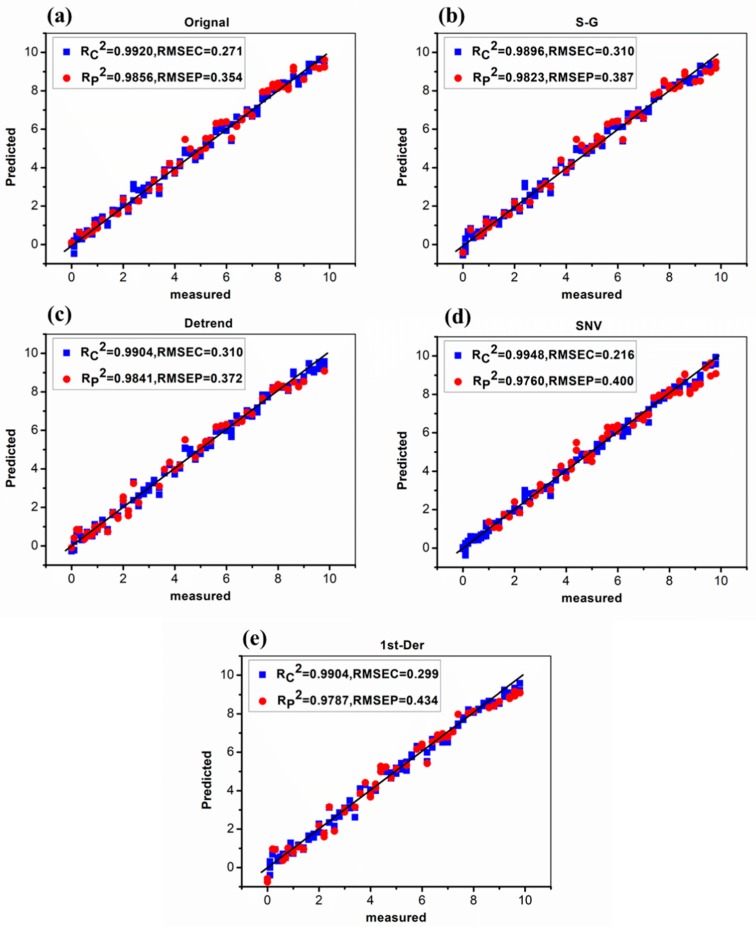
The model performance modeled by partial least squares (PLS) with different pretreatments. (**a**) Original; (**b**) Savitzky–Golay (S-G); (**c**) detrend (DT); (**d**) standard normal variation (SNV); (**e**) 1st-derivative (1st-Der).

**Table 1 molecules-24-01790-t001:** The proposed assignment of Raman peaks of SD.

Calculation (cm^−1^)	Solid (cm^−1^)	SERS (cm^−1^)	Assignments
543 (w)	553 (w)	552	υ carbonyl + δphenetole
633 (w)	647 (w)	647 (w)	υ carbonyl + δ phenetole+ υ (C-S) in Sulfamide
831 (w)	812 (w)	814 (m)	υ Pyrazole pyridine
927 (w)	926 (w)	922 (m)	δ (C-C) + υ (C-H) in Pyrazole pyridine group
1228 (m)	1232	1235	δ (C-H) in carbonyl
1308 (m)	1310 (w)	1305	δ (C-H) in ethyl
1398 (m)	1396 (m)	1401 (m)	δ (C-H) in methyl piperazine
1472 (m)	1487 (m)	1488 (m)	δ (C-H) in Pyrazole pyridine
1534 (vs)	1527 (s)	1530 (s)	δ (C-H) in Pyrazole pyridine
1580 (s)	1583 (vs)	1584 (vs)	δ (C-H) in Pyrazole pyridine

Note: vs = very strong; s = strong; m = medium; w = weak; υ = stretching; δ = deformable vibration.

**Table 2 molecules-24-01790-t002:** The precision and accuracy of method for the determination of sildenafil in cocktail.

Days	Added (mg/L)	Predicted (mg/L) Mean + SD	^a^ RSD (%)	Recovery (%)	Days	Added (mg/L)	Predicted (mg/L) Mean + SD	RSD (%)	Recovery (%)
Day one	0.3	0.307 ± 0.016	2.01	102.4	Day three	0.3	0.298 ± 0.012	1.53	99.6
	0.5	0.465 ± 0.045	5.59	93.0		0.5	0.469 ± 0.031	6.67	91.4
	0.7	0.641 ± 0.058	7.17	91.6		0.7	0.641 ± 0.058	7.17	91.6
	0.9	0.95 ± 0.071	8.71	105.6		0.9	0.93 ± 0.024	3.05	103.7
	3	3.09 ± 0.16	10.1	102.1		3	2.96 ± 0.12	11.9	98.7
	5	4.88 ± 0.090	10.9	97.7		5	4.84 ± 0.084	10.2	97
	7	6.59 ± 0.107	11.8	94.1		7	6.70 ± 0.047	5.8	95.8
	9	9.02 ± 0.13	16	100.2		9	8.96 ± 0.148	12.6	99.5
Day two	0.3	0.305 ± 0.013	1.59	101.8	Inter day	0.3	0.303 ± 0.015	1.54	101.2
	0.5	0.469 ± 0.036	4.8	93.9		0.5	0.468 ± 0.039	4.16	93.6
	0.7	0.640 ± 0.054	6.67	91.4		0.7	0.641 ± 0.055	6.92	98.8
	0.9	0.97 ± 0.066	8.09	108.1		0.9	0.952 ± 0.060	6.38	105.8
	3	3.07 ± 0.126	12.3	102.3		3	3.03 ± 0.13	11.8	101.2
	5	4.96 ± 0.102	12.5	99.2		5	4.89 ± 0.103	10.9	97.9
	7	6.74 ± 0.081	10	96.38		7	6.68 ± 0.105	11.24	95.4
	9	9.09 ± 0.103	12.7	101.1		9	9.03 ± 0.14	14.8	100.3

^a^ SD (standard deviation); RSD (relative standard deviation).

**Table 3 molecules-24-01790-t003:** The PLS modeling results based on 500–1700 cm^−1^ spectra under different pretreatments.

Pretreatment	Principal Components	Calibration		Prediction	
		^a^ Rc^2^	RMSEC	Rp^2^	RMSEP
Original	9	0.9920	0.271	0.9856	0.354
S-G	8	0.9896	0.310	0.9823	0.387
Detrend	8	0.9904	0.310	0.9841	0.372
SNV	10	0.9948	0.216	0.9760	0.400
1st-Der	8	0.9904	0.299	0.9787	0.434

^a^ Rc^2^ (the coefficient of determination of the calibration set); Rp^2^ (the coefficient of determination of the prediction set); RMSEC (root mean square error of the calibration set); RMSEP (root mean square error of the prediction set).

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
