# Peer review of "Rapid and Quantitative Determination of Sildenafil in Cocktail Based on Surface Enhanced Raman Spectroscopy"

_molecules, 2019, doi:10.3390/molecules24091790_

Reviewer 1 Report

The manuscript "Rapid and Quantitative determination of sildenafil in cocktail based on surface enhanced Raman spectroscopy", written by Lin et al., describes a new method for a determination of sildenafil using surface enhanced Raman spectroscopy. The theme is important, and the methodology seems to provide good results. However, method reliability and feasibility should be supported. I have following comments:

Line 110: "As seen in Figure 2, the SERS spectra of SD were basically consistent with the experiment-detected Raman spectra of SD and DFT-calculated Raman spectra of SD. which indicated that the SERS spectra of SD based on OTR 202 were feasible and reliable." I am convinced that calculated spectra are not the same as measured counterparts. There are many differences not only in the positions of spectral bands, but also in intensities. This statement should be rephrased. Also - similarity of calculated and measured spectra does not make the method reliable. Reliability has a clearly defined definition. What makes the spectrum feasible and reliable?

Numbers on X-axis in the figure 2-4 are not readable.

Line 165: "The results suggested that the predicted value was basically same as the true value, which indicated that the rapid detection of SD in cocktail by SERS was feasible and reliable". This statement has to be changed as well. It is very similar to my first point. "Basically same" is not a statistical statement. Similarity between predicted, and true values can not be described as reliability and feasibility. What about a long-term reproducibility of results? What about intra-day and inter-day statistics? All there parameters should be involved in the study.

The used SERS substrates should be described in better details. How was it applied?

The real sample should be better described.

Possible spectral interferences should be described and their effect discussed.

Author Response

Dear reviewer:

       Thank you for your comments and suggestions on our manuscript, and we have made changes a lot to our manuscript which you mentioned and suggested us re-writing. The following is our reply to your comments:

       1.Line 110: "As seen in Figure 2, the SERS spectra of SD were basically consistent with the experiment-detected Raman spectra of SD and DFT-calculated Raman spectra of SD. which indicated that the SERS spectra of SD based on OTR 202 were feasible and reliable." I am convinced that calculated spectra are not the same as measured counterparts. There are many differences not only in the positions of spectral bands, but also in intensities. This statement should be rephrased. Also - similarity of calculated and measured spectra does not make the method reliable. Reliability has a clearly defined definition. What makes the spectrum feasible and reliable?

       This sentence has been rewritten as follows “As seen in Figure 2, the positions of spectral bands and its intensities of SD SERS spectra were basically consistent with the experiment-detected Raman spectra of SD (Raman shifts < 5 cm−1), which indicated that the position of Raman peaks detected by SRES spectra of SD based on OTR 202 were feasible and reliable. Expect the differences between experiment-detected Raman spectra of SD and the DFT-calculated Raman spectra of SD 812 and 1487 cm-1, the DFT-calculated Raman spectra of SD were basically similar to the experiment-detected Raman spectra of SD whose Raman shifts (less than 10 cm−1) were within a reasonable range.

       2. Numbers on X-axis in the figure 2-4 are not readable.

       The sentence has been changed.

       3.Line 165: "The results suggested that the predicted value was basically same as the true value, which indicated that the rapid detection of SD in cocktail by SERS was feasible and reliable". This statement has to be changed as well. It is very similar to my first point. "Basically same" is not a statistical statement. Similarity between predicted, and true values can not be described as reliability and feasibility. What about a long-term reproducibility of results? What about intra-day and inter-day statistics? All there parameters should be involved in the study.

First of all, thank you very much for your comments and suggestions. The sentence has been revised to “According to Table 3, the intra and inter day RSD were less than 12.7 and 11.8%, respectively. And the intra and inter day accuracy% were ranged between (93.0–105.6%) and (93.6–105.8%), respectively. From the results, one can infer that precision and accuracy were within the acceptable limits”. And in order to obtain the long-term reproducibility of results, we did the experiment again, all the samples were detected by SERS based on OTR 202 of three consecutive days and then the linear regression equations at 1235, 1401, 1530 and 1584 cm-1 were used to predict the SD concentration in cocktail.

       4. The used SERS substrates should be described in better details. How was it applied?

       In this paper, the OTR 202 nanomaterials and OTR 103 produced by Opto Trace Technologies, Inc. (SuZhou, China). Since the colloid was purchased from the company, the preparation process and production process of the colloid were unknown. But in this experiment, to investigate the enhancement effects of Opto Trace Raman 202 (OTR 202), the structure, UV spectrum and Raman spectroscopy (RS) of OTR 202 were analyzed.

       5. The real sample should be better described.

       First, eight different SD concentration in cocktail (0.3, 0.5, 0.7, 0.9, 3, 5, 7and 9mg/L) were prepared. There were three samples of each concentration. Second, all the samples were detected by SERS based on OTR 202. Third, the linear regression equations at 1235, 1401, 1530 and 1584 cm-1 were used to pretreat the SD concentration in cocktail.

       6. Possible spectral interferences should be described and their effect discussed.

       The relevant discussion has been added in section 2.5 and 3.5.

       Thank you again for the comments and suggestions. If you have any questions, we hope you can contact us.

Best wishes

Lei Lin

Reviewer 2 Report

In this manuscript, SERS was used for the quantitative detection of sildenafil (SD) in cocktail. The SERS intensities against logarithm of the concentration shows lines, and thus the concentration of SD can be predicted. Moreover, the authors investigated the effect of the pretreatment. As a result, the original data without the pretreatment could be sufficient for the quantitative detection in the range of 0.1-10 mg/L. Thus, this manuscript can be publishable in the journal. 

However, Fig.3a shows the prominent peak at 810 cm-1 was observed from the cocktail even without SD. I wonder if ratios of the peak intensities at 1235, 1401, 1530, and 1584 cm-1 to those at 810 cm-1, which may be more robust than the original intensities, can be used for the quantitative detection. The authors can discuss the concentration dependence of the ratios. In Fig.3, the peaks at 1200-1600 cm-1 seem to be similar intensities despite the different concentration unlike Fig.4. Did you make these spectra all the same intensity? If so, the adjustment should be written in the caption. 

There are some small problems as follows. 

Line 16: The word “and” should be inserted after “Sildenafil (SD)”.

Line 29: The authors wrote as “R2p” here, while “Rp2” are written except for here. Which is the correct expression?

Line 62: sorbed -> adsorbed

Line 106: 1c -> 2c

Line 130: The word “absorbance” is not appropriate for SERS.

Line 144: It is written that “the Raman peaks … decreased gradually.” But, the statement may not apply to the intensities at the concentration < 2 mg/L, which seem to be drastically decreased. 

Fig.2-4: Numerals for the wavenumber are crowded onto the x-axes and thus are illegible. They may be better to locate on the x-axes at intervals of 200 cm-1

Table 4 (Lines 181-183): Although the authors explained the meaning of the words “Rc”, “RMSEC”, “Rp”, and “RMSEP” in the lines 253-255, they should be written also here because of the first appearance. 

Ref.18,24: The informations on the volume and pages may be missing. 

Ref.20,23,37: The informations on the pages are missing. 

Author Response

Dear reviewer:

       Thank you for your comments and suggestions on our manuscript, and we have made changes a lot to our manuscript which you mentioned and suggested us re-writing. The following is our reply to your comments:
       1. However, Fig.3a shows the prominent peak at 810 cm-1 was observed from the cocktail even without SD. I wonder if ratios of the peak intensities at 1235, 1401, 1530, and 1584 cm-1 to those at 810 cm-1, which may be more robust than the original intensities, can be used for the quantitative detection. The authors can discuss the concentration dependence of the ratios. In Fig.3, the peaks at 1200-1600 cm-1 seem to be similar intensities despite the different concentration unlike Fig.4. Did you make these spectra all the same intensity? If so, the adjustment should be written in the caption.

       The discussion has been added. It can be seen that there was faint SERS signal when the SD was 0 mg/L. This may be the SERS signal of some substances in cocktails. And the Y axis coordinates has been added in each spectrum in Figure 3.

2. Line 16: The word “and” should be inserted after “Sildenafil (SD)”.

It has been added.

3. Line 29: The authors wrote as “R2p” here, while “Rp2” are written except for here. Which is the correct expression?

The correct expression is Rp2 and it has been revised in the manuscript.

4. Line 62: sorbed -> adsorbed

It has been revised.

5. Line 106: 1c -> 2c

It has been revised.

6.Line 130: The word “absorbance” is not appropriate for SERS.

The word in this sentence has been deleted.

7. Line 144: It is written that “the Raman peaks … decreased gradually.” But, the statement may not apply to the intensities at the concentration < 2 mg/L, which seem to be drastically decreased.

The misrepresentation has been revised.

8. Fig.2-4: Numerals for the wavenumber are crowded onto the x-axes and thus are illegible. They may be better to locate on the x-axes at intervals of 200 cm-1.

It has been revised.

9. Table 4 (Lines 181-183): Although the authors explained the meaning of the words “Rc”, “RMSEC”, “Rp”, and “RMSEP” in the lines 253-255, they should be written also here because of the first appearance.

It has been added.

10. Ref.18,24: The information on the volume and pages may be missing.

It has been added.

11. Ref.20,23,37: The information on the pages are missing.

It has been added.

       Thank you again for the comments and suggestions. If you have any questions, we hope you can contact us.

Best wishes

Lei Lin

Round  2

Reviewer 1 Report

The scientific level of the revised version of the manuscript was considerably increased. I have no further comments / questions.

Author Response

Dear reviewer:

       Thanks for your reply!

Best wishes

Lei Lin

Reviewer 2 Report

The authors have revised the manuscript properly. 

In Fig.3, however, the top spectrum is still represented as not (f) but (d). It may be better to insert the vertical bars which shows the each intensities rather than to add the y-axis coordinates.

Ref.36: Are initials of their middle names missing?

Ref.37: The information on the page is still missing.

Author Response

Dear reviewer:

       Thank you for your comments and suggestions on our manuscript, and we have made changes a lot to our manuscript which you mentioned. The following is our reply to your comments:

1.       In Fig.3, however, the top spectrum is still represented as not (f) but (d). It may be better to insert the vertical bars which shows the each intensities rather than to add the y-axis coordinates.

Figure 3 has been revised now.

2.       Ref.36: Are initials of their middle names missing?

The mistake has been corrected.

3.       Ref.37: The information on the page is still missing.

The information on the page has been added.

Thank you again for the comments and suggestions. If you have any questions, we hope you can contact us.

Best wishes

Lei Lin